# Will You Become the Next Troll? A Computational Mechanics Approach to the Contagion of Trolling Behavior

**DOI:** 10.3390/e27050542

**Published:** 2025-05-21

**Authors:** Qiusi Sun, Martin Hilbert

**Affiliations:** 1School of Information Studies, Syracuse University, Syracuse, NY 13244, USA; 2Department of Communication, University of California, Davis, CA 95616, USA; hilbert@ucdavis.edu

**Keywords:** complex systems, predictive models, social media, trolling

## Abstract

Trolling behavior is not simply a result of ‘bad actors’, an individual trait, or a linguistic phenomenon, but emerges from complex contagious social dynamics. This study uses formal concepts from information theory and complexity science to study it as such. The data comprised over 13 million Reddit comments, which were classified as troll or non-troll messages using the BERT model, fine-tuned with a human coding set. We derive the unique, minimally complex, and maximally predictive model from statistical mechanics, i.e., ε-machines and transducers, and can distinguish which aspects of trolling behaviors are both self-motivated and socially induced. While the vast majority of self-driven dynamics are like flipping a coin (86.3%), when social contagion is considered, most users (95.6%) show complex hidden multiple-state patterns. Within this complexity, trolling follows predictable transitions, with, for example, a 76% probability of remaining in a trolling state once it is reached. We find that replying to a trolling comment significantly increases the likelihood of switching to a trolling state or staying in it (72%). Besides being a showcase for the use of information-theoretic measures from dynamic systems theory to conceptualize human dynamics, our findings suggest that users and platform designers should go beyond calling out and removing trolls, but foster and design environments that discourage the dynamics leading to the emergence of trolling behavior.

## 1. Introduction

As human interactions in digital environments increasingly became an important part of our daily activity, the concerns about misconduct on online platforms have increased. According to a national survey in 2023 [1], 33% of adults in the U.S. have experienced any form of online deviant behavior. Among those who have experienced deviant behaviors, 45% attributed it to their political views, and 75% encountered it on social media. One of the deviant behaviors that draws the most attention is called trolling behavior. Trolling is considered a deviant behavior that diffuses misinformation [2], provokes emotional responses, or disrupts on-topic discussions [3]. The motivations of the behavior vary from simple attention attractions [4] to well-calculated manipulations of public opinions [5]. In addition, the effects and consequences of the behavior also vary across audiences and contexts, leading to significant problems that affect individuals and online communities.

Much of the work on trolling behavior has focused on its language aspect, investigating the identification and antecedents of trolling behaviors [5,6,7,8]. Scholars have largely treated trolling as an individual linguistic characteristic, believing that capturing the features of messages would solve the puzzle of “what”, “how”, and “why”. The underlying premise has been that if someone talks like a troll, they are a troll. However, this perspective overlooks the complexity of trolling as an online deviant behavior. In addition to linguistic approaches, other scholars have examined trolling through sociological, psychological, and interactional lenses, emphasizing contextual, relational, and identity-based dimensions of the behavior [7,9]. These studies highlight that trolling cannot be fully understood without considering the broader social dynamics in which it occurs. Therefore, our approach goes beyond linguistic markers, positing that if someone behaves like a troll, regardless of individual characteristics, they are a troll. In this study, we define trolling as a behavior to attract responses and gain influence by making time-wasting comments on controversial topics, and using provocative language and strategies, such as referring to a person, repeatedly commenting, and using misinformation.

Per definition, behavior is a dynamic process. However, traditional research in information science and communication has largely overlooked the temporal dynamics of behaviors as they unfold over time [10]. In this study, we aimed to fill this gap by adopting a dynamic behavioral perspective to deconstruct the communication processes of trolling. With traceable digital footprints from Reddit in 2016 and 2017 and an autoregressive model-based investigation into human–human interactions, we represent new opportunities to examine whether and how trolling behavior is contagious on a large scale. We use the so-called predictive state model [11] from statistical mechanics, which helps to identify the unique, minimally complex, maximally predictive model of the dynamics. By examining how trolling behaviors have evolved and interacted with social factors, we have provided a more comprehensive understanding of the mechanisms driving this complex online deviant behavior.

## 2. Trolling as a Dynamic Communication Process

From an information-theoretical perspective, communication behaviors have been studied as dynamic systems [12,13,14] where the influence of communication actors is dynamically embedded. From such a perspective, information is viewed as the resolution of uncertainty [15], providing a method to quantify the predictability of dynamic communication systems. As a complex dynamical system, an individual’s communication behaviors are considered discrete components or constituents that are part of a collective, in which each component or constituent will affect one another to create a pattern, an order, or a structure at the level of the collective that cannot be observed on the individual level [16]. Furthermore, communication behaviors generally involve more than one individual, so the dynamic of communication behaviors is likely to involve other individuals, contingent on the ongoing and simultaneous flow of communicative actions from both individuals involved, creating patterns, orders, and structures. In such a co-dynamic system, individuals’ behaviors are dynamically altered by their social partners.

Trolling is often defined through textual or contextual perspectives, with a substantial body of literature focusing on the linguistic content of messages [17,18]. However, scholars have yet to reach a conclusion on a universal definition of trolling; rather, the previous literature defined trolls in various ways. Some focused on the deceptive nature of trolling messages [5,19,20], claiming that trolls use fake personas and misleading messages to impose opinions on others, damaging trust in online communities and finding valuable information from others. Others argued that provocation is the main feature of trolling [7,21,22,23], where trolls use erroneous or inflammatory information to trigger negative reactions from ordinary Internet users. In addition, trolls are defined differently depending on context. For example, in gaming platforms, trolling is a clear concept associated with the griefing culture [24] and is treated as a strategy for competition. In political discussions, trolls are generally considered ideological devices that express extreme political opinions and mislead their audience [25]. The non-unified definition of trolling indicates that trolling is a complicated multidimensional concept that cannot be simplified as a linguistic aspect of messages. Thus, in this study, we approach from a comprehensive perspective how trolling is a behavior with aggressive characteristics and a misleading nature, echoing the inclusive definition of Cheng, Bernstein [26] where trolling is behavior that “falls outside the acceptable bounds” [26] and the definition of Herring, Job-Sluder [4] focused on both the behavior’s characteristics and outcome.

Acknowledging the early roots of trolling research in personality psychology, prior studies have shown that individuals who engage in trolling behaviors often exhibit distinct traits and personality profiles [7,27,28]. Trolling behavior could be recurring, reflecting individuals’ personalities (e.g., the Dark Tetrad [27,28]), biological traits (e.g., low baseline arousal [29]), or operational mechanisms developed by acculturation to Internet culture. However, when individuals read, comment on others’ posts, and reply to others’ comments, they may be influenced by the contextual environment or even by the one they interacted with. The provocative nature of trolling behaviors may trigger mood swings. Individuals may experience negative emotions and further engage in trolling behaviors to outlet emotions and enact revenge. Experiencing previous trolling behaviors may raise the likelihood of an individual engaging in future trolling behaviors. Vice versa, it is also possible that angry individuals who post trolling messages will calm down after reading some non-trolling comments. Thus, trolling can be an induced behavior, echoing some scholars’ hypothesis that ordinary people can be triggered into engaging in trolling behaviors by discussion content and sentiments [26,30].

We adopt the approach of information theory [31,32], specifically an information-theoretic approach to dynamic systems theory [33,34,35]. Just like Shannon [35] conceptualizing the sequential letters of the alphabet of the English language as dynamic [35], the trolling dynamic system can be chronologically viewed as a temporal sequence. Thus, whether and how individuals engage in trolling behaviors can be understood as related to the co-influence between trolling and non-trolling behaviors, and social influence can be modeled with parallel sequence data. Based on the temporal sequence of trolling and non-trolling behaviors and parallel social trolling and non-trolling behaviors from others that (potentially) interact with each other, we can calculate the unique, minimally complex, maximally predictive model of the dynamic, known as the predictive state model [36,37,38]. It reveals the complexity of the dynamic and gives an optimized representation of it, allowing us to study its intrinsic properties.

According to previous research on behavior patterns that used this approach [39], such a temporal sequence can be analyzed as an autoregression in which their own past behavior influences their future behavior. In the literature of computational mechanics, the respective information-theoretic model is known as the ‘ϵ−machine’ (say ‘epsilon machine’) [40]. Another possibility is that users are affected by the contextual environment, where they react to others instead of posting independently. The literature of computational mechanics refers to this as an ‘ϵ−transducer’ [39,41]. If so, the dynamical characteristics of the users’ temporal sequence should reflect such relative dependence, where a user’s temporal sequence will be associated with another user’s temporal sequence. Furthermore, users’ behavior may be much more complex, with multiple influences, and their own history and the contextual environment can jointly impact their future behaviors. To this end, we seek to compare the relative influence of users’ prior behaviors and the social inputs they receive on the emergence of trolling. Thus, we first ask the following question.

RQ1: Is trolling behavior socially induced?

Social contagion is defined as the spread of affect, attitude, or behavior from one person to another [42], which emphasizes that attitudes and behaviors spread like epidemics, and people need to come in contact with others who have already adopted the attitudes or behaviors. However, due to technological limitations, studies on contagion tend to be limited to experimental settings [43,44] or conducted on a macro level, treating it as a fundamental force that drives the formation and propagation of psychological states (e.g., emotions [45]), opinions and attitudes [46], and behaviors [47], which is easily confused with other similar models, namely, social influence and social learning [48,49]. From an information-theoretic perspective, we can zoom in on individual behavioral dynamics, providing an opportunity to investigate the direct influence of contact on an individual level.

Previous research has shown that humans instinctively align the emotional states they receive during communication interactions [50]. Emotional contagion is reflected in such facial, vocal, or postural alignment, as well as similar neurophysiological and neurological reactions [45]. Moreover, emotional contagion is also shown in computer-mediated communication [49]. For instance, in 2014, the controversial Facebook emotion experiment [51] demonstrated that emotions can be increasingly intense during the spread on social media, where individuals can experience similar emotions when reading emotional social media posts, resulting in similar emotional expressions. As a response to emotional contagion, individuals will show behavioral synchrony [45], leading to behavioral contagion. In our case, the contagious negative emotions may increase the probability of individuals engaging in trolling behaviors. With intensified emotions, trolling behaviors may persist across affected people and spread further.

In addition, previous research also investigated the contagion of deviant behaviors with the mechanisms of generalized reciprocity and third-party influence. Generalized reciprocity refers to the victims recouping by “paying it forward [43]”, while third-party influence is mimicry after observing the behavior of others. It echoes the hypothetical normalization theory [52,53,54] that deviant behaviors may not be taken personally by victims and observers but rather attributed to a societal pattern instead. Empirical research has also provided evidence for the contagious nature of trolling behaviors. For instance, Shen, Sun [55] finds that exposure to toxic behavior in massively multiplayer online games increases the likelihood of players becoming toxic, while Cook, Conijn [9] suggests that trolling may follow a rapid contagion cycle with victims and bystanders adopting trolling behaviors. From a dynamic system perspective, such a phenomenon can be described by the dynamic characteristics of individual interactions. The temporal sequence of an individual’s trolling behavior may show a pattern in which the probability of trolling behavior is increased with trolling behavior from contextual sequences. In other words, we examine whether prior trolling behavior by other users influences an individual’s likelihood of engaging in trolling themselves. As such, trolling behavior could be contagious and spread on social networks through interaction and discussion. Thus, we ask the following question.

RQ2: Is trolling behavior contagious?

### Predictive State Models

The computational mechanics approach allows us to discern the hidden mechanism of how the information is stored, processed, and transformed over time, constructing a unique model that maximizes the predictive power but minimizes the complexity for the discrete-state, discrete-time stochastic process to describe the dynamical characteristics of temporal sequences of human behaviors. From a mathematical perspective, it is a sufficient statistic. From a social science perspective, it allows us to detect a unique model for behavioral patterns, look at the mathematical architecture of behavior’s temporal sequences, and decompose the procedure for generating sequences.

Specifically, the two predictive state models we employ are unifilar hidden Markov models, one with a single process (also known as “ϵ−machine”) and the other with an input–output process (also known as “ϵ−transducer”). Its hidden states consist of “a set of histories, all of which lead to the same set of futures. It’s a simple dictum: Do not distinguish histories that lead to the same predictions of the future [37]”. Mathematically, the predictive statistic of the past for predicting the future of a conditionally stationary stochastic process is a minimal sufficient statistic for prediction. Thus, for each possible predictive distribution, we could find a class of pasts that induce this predictive distribution and a statistic ϵ that can map a past into an equivalence class for that past. Here, the equivalent is defined as “if two pasts result in statistically equivalent future, they are equivalent [39]”. The states of a unifilar hidden Markov model represent a partition for all pasts based on the conditional futures they induce.

Simply put, what we observe in a temporal sequence is a representation of some hidden states, which can be considered as a piece of memory containing what the system will do when receiving an input. Usually, those states are not the individual events we can observe but are associated with a combination of events. In our case, the hidden states of trolling behaviors can be psychological, emotional, or other possible states that we may not be able to name without further investigation, for example, the states of anger or not. When individuals are angry, the likelihood of trolling may increase, but it does not guarantee that individuals will engage in trolling behaviors. The angry state does not distinctly map to trolling behaviors. We can observe the trolling behaviors but not the hidden anger state.

Investigating hidden states is crucial for advancing our understanding of trolling behaviors, as it allows us to move beyond surface-level correlations and explore how users enter, are attracted to, or disengage from trolling states. By reconstructing the sequences of user behavior, we aim to detect whether trolling follows distinct and recurring states, such as sustained trolling, transitional phases, or reversion to non-trolling. These patterns, while not directly observable, may reflect stable behavioral mechanisms that help explain why trolling behavior can escalate or persist, even without immediate provocation. Furthermore, hidden states can serve as a conceptual bridge between neuroscience and psychology: they align with neurobiological mechanisms (e.g., emotional regulation in the brain) while also resonating with psychological frameworks of motivation and memory. Hence, we ask the following:

RQ3: Does trolling follow identifiable cyclic behaviors and attractor states, or patterns of dispersion in the state space?

The first model is a univariate predictive state model. It is an autoregressive model [38] that quantifies the minimal size and optimally predictive power model to predict individuals’ behaviors based on their past. Illustrated in Figure 1a, it assumes that an individual’s future behavior is only influenced by their own past behaviors, so we also call it a self-driven model. In self-driven models, the probability of whether an individual trolls or not in the future time is determined by whether they troll in time instants before; that is to say, the probability of whether an individual trolls or not in the future time instant t is determined by whether they are trolls in time instants before t.

Practically, at any given time instant t, an individual either troll or not, which can be denoted by $X_{t}$. There are only two statuses for trolling behavior $\;X_{t},\;X_{t}=1\;or\;0$, where 1 is assigned to $X_{t}$, when the individual’s behavior is identified as trolling behavior, and 0 otherwise. We assume that the immediate past $X_{t-1}$ is the past behavior that can influence the future behavior; then, the probability that the individual has a behavior $x_{t}$ at time t on the condition of that individual having the behavior $x_{t-1}\;$ at time t−1 is as follows:$$P\left(X_{t}=x_{t}|X_{t-1}=x_{t-1}\right)$$

The second model expands the autoregressive logic to include a second parallel time series that is considered to influence the ongoing dynamic denoted in Figure 1b. Following the modeling approach of computational mechanics, if the ϵ−machine is a finite state machine that predicts the individual’s future behavior based on its past behavior, we now consider an input–output transducer that predicts the individual’s future behavior based on its past and the influence of an external source [56]. In our case, we refer to it as the social-induced model. It assumes that an individual’s future behavior is influenced by their past behavior and the past behavior of people they interacted with. It aims at capturing social influence and contagion. Similarly to the self-driven model, we derive the unique model with minimal complexity and maximal prediction power for an individual’s behavior. In addition, equivalence classes are identified over joint self and social pasts, and a mapping partition from the current joint past to its equivalence class is detected. Thus, additionally to the individual’s behavior $X_{t}$, we introduce a new variable of social inputs $Y_{t}$, and in our case, $Y_{t}$ indicates the parent comment.$$Y_{t}=\left\{\begin{array}{c} 1,\;\;\;if\;the\;parent\;comment\;is\;trolling\\ 0,\;\;\;otherwise \end{array}\right.$$

Then, we have the following:$$P\;(X_{t}=x_{t}|X_{t-1}=x_{t-1},Y_{t-1}=y_{t-1})$$

## 3. Methods

### 3.1. Data Collection

The data used to feed into stochastic models are obtained from the Reddit comment history available from Google Cloud BigQuery (a fully managed, serverless, petabyte-scale data warehouse service offered by Google Cloud Platform). Based on a recent report, Reddit is the 16th most visited website globally, with more than 430 million monthly active users [57]. Reddit has subcommunities designated for specific topics called sub-reddits, where people can post, comment, and react to others’ posts and comments. Individuals can subscribe to any sub-reddits to enjoy different content and discussions (while some sub-reddits may have restricted rules for admitting subscribers). One thing worth noting is that Reddit encourages users to stay anonymous, unlike Facebook or Twitter. It protects users’ identities and does not require any identity verification [58]. This policy, on the one hand, supports the freedom of expression [58], but on the other hand, it also fuels more uninhibited trolling, toxicity, or hate speech on the platform. Even with Reddit’s anti-harassment policy, we observed trolling behaviors on the platform.

We chose to work with the time period from 2016 to 2017, when there was a strong social push for politically cross-cutting discussion. We identified the 1000 most active accounts on Reddit based on the number of comments posted during the time period. The comments from those active accounts and the comments they directly replied to, which we considered parent comments, from 2016 to 2017, were retrieved from Google BigQuery. If a comment replied to a post, we also retrieved the post as the parent comment. We obtained a dataset with 17,495,543 comments from 1000 accounts with all their parent comments. Then, one human annotator worked with sample comments from each account to identify Automods, Reddit’s automatic tools to reinforce sub-reddit rules. The Automods were excluded from our dataset, leaving 826 users with a total of 13,456,759 comments, with a maximum of 89,763 comments and a minimum of 9642 comments (*M* = 16,174.89, *SD* = 9477.37).

### 3.2. Data Annotation and Classification

Seven undergraduate students worked as human annotators for the categorization of troll messages. The target user’s comments and parent comments were labeled “trolling message” and “non-trolling message”, respectively, 1 and 0. A coding manual was developed by the annotators based on the definition of Herring, Job-Sluder [4]. Trolling was identified as comments that satisfied the following criteria: (1) using emotionally provocative language, (2) referring to a person or a group, (3) containing incorrect information, or (4) containing unexpected or unrelated messages. Annotators went through 5 rounds of training and discrepancy solving, reaching inter-coder reliability in a Fleiss’s kappa of 0.656, which is intermediate to good, according to Fleiss, Levin [59]. They then moved on to independent annotation. Including the annotation training sets, they annotated 15,558 (0.12%) comments for supervised model training. During the coding process, annotators also encountered deleted and removed comments. Unlike previous research [60,61], where deleted and removed comments were excluded from analysis, in this study, we kept those comments. As we analyzed the individuals’ behaviors as time series, excluding time instants from a time series will result in an incomplete time series and misrepresent the mathematical structures, so keeping the “missing value” in the analysis is essential for model building. Moreover, deleted comments were taken down by users themselves because they might regret what had been said, the comments have been downvoted, or the comments received no replies, but removed comments were taken down by platform moderators, usually because of a violation of Reddit or sub-reddit policy. That is to say, removed comments are more likely to be trolling messages, but deleted comments may not necessarily be related to trolling behaviors. We coded the deleted comments as non-trolling messages and removed comments as trolling messages.

With the large dataset, we adapted the incivility BERT model of Davidson, Sun [62] for automatically identifying trolling. Even though the model was not built for the purpose of trolling identification, it is beneficial when adapting for our dataset because the model used BERT technic to pre-train for domain adaptation on 3 million unlabeled Reddit comments using a masked language modeling objective, which helped capture the Reddit culture. Prior to applying the model, we conducted minimal text preprocessing to preserve the naturalistic structure of the data, which included normalizing whitespace and removing special characters, emojis, and URLs. Instead of using incivility annotated data, the model was then fine-tuned with four epochs on 10,000 comments from the human-annotated trolling comments, with 10% set aside for model validation. The other 5558 annotated comments were then used for model testing. The model reached a final *F-1* score of 0.76.

### 3.3. Predictive State Model Building

As stated above, we follow the practice in behavioral science that reduces the communication flow into a probabilistic sequence. Because the self-driven and social-induced models assume that users’ online behavior can be modeled as a conditionally stationary stochastic process, the distribution over futures is independent of the time index conditional on the observed past. To approximate the assumption, we considered each comment as a unique incident happening at a temporal point, and we then had a series of discrete incidents as a temporal sequence of trolling behaviors for each Reddit active user for a two-year time period. Similarly, we also generated a two-year temporal sequence of parent trolling behaviors for each Reddit active user from the parent comments. It is important to note that our analysis did not consider time stamps; we simply note that one event happens after another in the tradition of dynamical system sequence data. Though we acknowledge that for individual users, a lot of things may happen between two comments, and the time intervals may also vary, we have a long time period that are likely to even out the day-to-day differences. In addition, the parallel comparison between self-driven model and social-induced model focused on the difference on predictive power introduced by social input, thus, we do not focus on the variance of reaction time.

For both the ϵ−machine and ϵ−transducer models, we used the Causal State Splitting Reconstruction (CSSR) algorithm [39] to infer the models from individual users’ time series. The process could be visualized as using a sliding window with a certain length L to move across the temporal sequence and record the frequencies of different combinations of subsequences. The CSSR algorithm works in two phases: in the first phase, it determines a set of weakly prescient states, and in the second phase, it removes transients and splits the causal states. In this algorithm, $L_{max}$, the maximum history length, which is the length of the sliding-window, is used for determining the candidate causal states in the first phase, and size α is the control for the probability of splitting causal states that indirectly controls the total number of the causal states detected by the model in phase two. To achieve an $L_{max}$ with a minimum log loss, a train/test split method was used, where half of the data were used to estimate the ϵ−machine with a specific $L_{max}$, and the other half was used to determine how well the estimated ϵ−machine can describe the data. Then, the optimized $L_{max}$ was used for machine estimation. Considering that the majority of the comments do not reply to the same parent post, we distinguish among predictive states if they have a similarity less than α = 0.001.

### 3.4. Measurements

Specifically, we calculated three complementary information-theoretic measures as we constructed predictive state models, namely, predictable information E, predictive complexity C, and remaining uncertainty h, illustrated in Figure 2 [63]. The Venn diagram presents past and future information and the communication from the past to the future regarding entropy H. In information theory, entropy measures the uncertainty of an event based on its probability distribution. Together with the number of states and the maximum history length $L_{max}$, these measures were used to inform our hypotheses, following their insightful application in other social science dynamics [64,65,66].

#### 3.4.1. Predictable Information

The previous literature named predictable information E differently as “effective measure complexity” [67], “excess entropy” [68], or “predictive information” [69]. It refers to the mutual information [15] between the past and the future, which is the amount of information that the past communicates to the future. The Venn diagram (Figure 2) presents it as the overlapping area of the green and red circles. The higher the predictable information is, the more subsequences from the past are used to predict the future.

#### 3.4.2. Predictive Complexity

The predictive complexity C quantifies the minimum amount of information that is required for a process to communicate all predictable information (E) from the past to the future [63]. To optimally predict a process, C is the amount of stored information needed. As both the ϵ−machine and ϵ−transducer models are the unifilar hidden Markov models of the dynamic with the minimum size but maximum predictive power, predictive complexity measures the size of the hidden Markov models, which is showed as the orange dashed ellipse in Figure 2. If we consider E the capacity of effective information transmission of a communication process, then C is the sophistication of it. The larger C is, the more complex the process is.

#### 3.4.3. Remaining Uncertainty

As shown in Figure 2, the conditional entropy $H\left(future\right|\;past)$ is the remaining area of the red circle after subtracting the overlapped area with the green circle, which is the amount of uncertainty left about the future after using the past to predict the future. It quantifies the information needed for predicting the future in addition to what can be informed from the past, possibly including information from the outside of the system, channel noise, or measurement error. We use the per-symbol rate h to indicate the remaining uncertainty, which is the conditional entropy scaled with the window length L. A larger h means a higher probability of prediction error; in other words, a higher remaining uncertainty indicates a larger prediction error, and a more unpredictable future. In addition, in our predictive model, as we stated in the previous section, the window length is used as the maximum history length $L_{max}$, and the remaining uncertainty rate h is calculated as $H\left(future\right|\;past)$ scaled with $L_{max}$.

#### 3.4.4. Number of States

The number of states indicates the size of the machine. A larger machine with more states has more possible transitions and more potential uncertainty about which state predicts the next future event. Thus, a larger machine requires more information to determine both the state of the process and where it moves. In other words, a high number of states implies a complex machine. An extreme case is a one-state machine, where the dynamics might be completely random or completely deterministic. There is just no pattern in such a machine. For instance, a random coin flip has one single state, as well as a dynamic that always produces the same. The number of states is a simplified approximation of predictive complexity C, which is the entropy of those states (i.e., weighted by their probability of occurrence).

#### 3.4.5. Maximum History Length

As mentioned above, the maximum history length is the optimized length of the sliding window for determining the causal states. It indicates a long enough history length to sufficiently predict future behavior. Thus, a machine with a larger maximum history length requires more information for state determination. It is used in the model-building process to balance the complexity and accuracy of a machine. In our analysis, it was treated as an indicator of complexity.

## 4. Results

### 4.1. Self-Driven Models

We first explored ϵ−machine architectures across the users. Among the 826 users we tested, 713 (86.32%) users only have one causal state, followed by 35 (4.27%) users with three causal states and 21 (2.56%) users with four states. The largest number of states detected from our dataset is eight. The full distribution of states is shown in Figure 3a. The number of causal states in ϵ−machine provides a rough reflection of the complexity of the user’s behavior because each causal state is a “further refinement of the past for predictive sufficiency [39]”. The majority of the users had only one state, indicating that their trolling behavior is like flipping a coin that, whether troll or not, is not a decision made based on their own past behavior(s). For those users, the state of troll or not troll is independent. Whether they acted as a troll is random or, at least, for a reason that cannot be detected from the self-driven model. There is no obvious pattern detected from their own behaviors, and self-driven trolling behavior is not complex. Figure 3b presents the distribution of the maximum history lengths used for the estimated models, showing 455 (55.13%) users had a maximum history length of one, 212 (25.64%) users had a maximum length of four, and 67 (8.12%) users had a maximum length of five. The maximum history length suggests how many past behaviors have been used to predict future behaviors. Thus, based on the maximum lengths of the ϵ−machines, more than half of the users only remembered the instant past behavior.

Secondly, we investigated the complementary information-theoretic measures of the predictive state models. Figure 4a,b show the relations between predictive complexity C, remaining uncertainty h, and predictable information E after removing outliers and the leftmost data points (C *=* 0). Unsurprisingly, both remaining uncertainty h (*r*(107) = 0.547, *p* < 0.001) and predictable information E (*r*(107) = 0.7, *p* < 0.001) are highly correlated with predictive complexity C. Models that require more stored information for prediction are more likely to have a high rate of remaining uncertainty and high predictable information. In short, more complex processes communicate more from the past to the future, but at the same time, the probability of prediction error is also higher.

Moreover, ANOVA tests were run to further explain the architecture of self-driven models between the information-theoretic measures, the number of states, and the maximum history length $L_{max}$. Specifically, ANOVA tests were run on models with more than one state since a one-state self-driven model basically follows a binomial distribution where the past and the future are independent and share no information. Presented in Table 1, the results showed that models with more states are more likely to have larger predictive complexity and remaining uncertainty, and models with a longer history length are more likely to have higher remaining uncertainty. This echoed the investigation of information-theoretic measures that showed a complex self-driven model with a long memory is more unpredictable [65,66]. However, post hoc analyses using Tukey’s honestly significant difference indicated that there is no statistical difference among predictive complexity and remaining uncertainty of five-, six-, seven-, and eight-state machines, showing that with the increasing number of states, the machine did not give us more information after five states. That is to say, self-driven models with more than five states did not explain more of the trolling behaviors.

### 4.2. Social-Induced Models

We also derived ϵ−transducer models with parent time-series inputs, resulting in a majority of models with two causal states (374 users, 45.32%) or three causal states (350 users, 42.43%). The largest number of states detected from the models was eleven (seven users, 0.85%). The distribution is shown in Figure 5a. Most users’ trolling behaviors can be described by a two-state or three-state ϵ−transducers, suggesting a hidden pattern guiding most users to troll or not troll. Furthermore, the hidden patterns are more complex and difficult to understand for some other users with more states. In addition, Figure 5b shows the distribution of maximum history length for ϵ−transducers. Similarly to self-driven models, around half of the users (401 users, 48.54%) had a maximum history length of one, followed by the maximum length of three with 233 users (28.21%) and the maximum length of five with 87 users (10.53%). Again, half of the users only remembered the instant past interaction with others, and the other half might have a better memory that can remember more interactions.

The complementary information-theoretic measures of the predictive state models were investigated. Figure 6a,b show the relations between predictive complexity C, remaining uncertainty h, and predictable information E after removing outlier and the leftmost data points (C = 0). Remaining uncertainty h (*r*(785) = 0.371, *p* < 0.001) and predictable information E (*r*(785) = 0.309, *p* < 0.001) are both significantly correlated with predictive complexity C. Similarly to self-driven models, more complex social-induced models are more likely to have a high rate of remaining uncertainty and more predictable information communicated from the past to the future.

To further understand the structure of social-induced models between the information-theoretic measures and the number of states and the maximum history length $L_{max}$, ANOVA tests were employed. Presented in Table 2, the results showed that the number of states and maximum history length of the models are significantly associated with the information-theoretic measures. Models with more states and a longer history length are more likely to have larger predictive complexity, predictable information, and remaining uncertainty, which indicates that a complex model with long memory provides more information for prediction but, at the same time, is more unpredictable. Similarly, the post hoc analyses show that there is no statistical difference among predictive complexity, predictable information, and remaining uncertainty of machines with five to eleven states, indicating that machines with more than five states are too complicated to provide more information for behavioral prediction.

### 4.3. Self-Driven vs. Social-Induced Models

To address RQ1, we applied a Welch’s *t*-test showing a significant difference in the number of states between the social-induced and self-driven models (*t* = 7.947, *df* = 824, *p* < 0.0001). The social-induced models had significantly more states than the self-driven models, showing a more complicated structure of the trolling process when involving social inputs. In addition, a Welch’s *t*-test was applied to the predictable information E between self-driven and social-induced models (*t* = −5.615, *df* = 824, *p* < 0.0001), indicating that social-induced models provide significantly more information for predicting the future. The social-induced models provide additional information about users’ trolling behaviors compared with the self-driven models in general. Although it seems that trolling behaviors are haphazardly based on their own previous behaviors for most users, the social-induced models reveal a complex hidden pattern guiding the behavior. Both self and social inputs provide information to predict future behaviors, but the social-induced model describes the trolling process better. Trolling behavior is self-social-together-motivated.

RQ2 asked whether the trolling behavior was contagious or not. Testing contagion on an individual level involved testing whether a user would adopt trolling behavior after encountering a troll. In our analysis, that is, this looked at whether a trolling parent comment will lead to a trolling comment. As in the social-induced models, one state suggests that trolling behavior is more likely to be random, while multiple states indicate hidden patterns behind the behaviors. A Welch’s *t*-test was applied for the number of trolling messages between users with a multiple-state social-induced model and a one-state social-induced model, and the result (*t* = 0.717, *df* = 824, *p* < 0.05) shows that users with hidden patterns are more likely to troll, supporting the claim that users are socially influenced by previous trolling messages from others.

### 4.4. Average Machines

To answer RQ3 and better illustrate the dynamic pattern of multiple-predictive-state self-driven and social-induced models, we also investigated average machines with different states. There are 109 (13.25%) users with a multiple-state self-driven model and 790 (95.57%) users with a multiple-state social-induced model. We first applied a Welch’s *t*-test for the predictive complexity between self-driven models and social-induced models, and the result (*t* = 3.883, *df* = 824, *p* < 0.001) supports that trolling behavior is a socially influenced complex dynamic with sophisticated hidden structures. An exploratory test of the number of comments between one-state social-induced models and multiple-state social-induced models showed that activeness was not related to the dynamic patterns of trolling behaviors (*t* = 0.381, *df* = 824, *p* = 0.704).

We then take the average of 4 two-state ϵ−machines, 35 three-state ϵ−machines, 374 two-state ϵ−transducers, and 350 three-state ϵ−transducers. To construct the average machines and transducers, we first aligned the states across models with the same number of hidden states, since hidden states are unlabeled and permutation-invariant. We arbitrarily chose one model as the reference model and aligned all others to it. For each remaining model, we computed a cost matrix that quantifies the distance between its transition probability vectors and those of the reference model. Then, the Hungarian algorithm [70,71] was used to find the optimal permutation of states that minimizes the total cost, and the transition and emission matrices of each model were reordered accordingly. Once all models were aligned to the same state ordering, we calculated the element-wise average of the aligned transition and emission matrices to derive a representative average model that captures the shared structure across the individual models.

Figure 7a shows the average ϵ−machine for the four two-state users. Their behavioral pattern distinguishes between two states, which we call the troll state and the non-troll state, and they process memory for both stages: as they switched from non-troll to troll (1), they were more likely to remain troll (1: 67%), and vice versa (0: 69%). Meanwhile, the probability of switching from troll to non-troll (0: 33%) and that of switching from non-troll to troll (1: 31%) are similarly lower.

In addition, the average three-state users’ ϵ−machine is illustrated in Figure 7b. There is a stage in addition to the troll and non-troll states; we call it the transition state. With a 39% chance, this user will advance to the troll state from the transition state and a there is a 61% chance they will advance to the non-troll state. Once the user arrives at the troll state, they will stay there with a probability of 76%. Note that the transition state cannot be reached directly from the troll state, and the troll state cannot be directly reached from the non-troll state. It seems that the transition state serves the purpose of a threshold for becoming a troll or not. When in a non-trolling state, the probability of staying there is quite high, 68%. When in the trolling state, the probability of staying a troll is even higher, 76%. Being in the transition state, the user seems to be in an uncertain and unstable situation of becoming a troll or not.

Moving to the average predictive state models that incorporate social inputs, the knowledge of the recent past of both their own and their social input (parent comment) behaviors provides sufficient information for predicting their future behavior. Figure 8a highlights the average two-state ϵ−transducer. Starting in the non-troll state, if the user is replying to a troll parent comment, the likelihood of switching to the troll state is much higher (1|1: 48%) than replying to a non-troll parent comment (1|0: 28%). Once a user reaches the troll state, replying to a troll parent comment will reinforce the stay in that state (1|1: 72%). Similarly, once a user switched to the non-troll state, in which one is more likely to respond to a non-troll comment (0|0: 0.45 vs. 0|1: 0.28), replying to a non-troll parent comment will also reinforce the stay in that state (0|0: 71%).

Figure 8b is the average ϵ−transducer for users with three states. The user exhibits both self and social memory in the sense captured by the model. Unlike the average three-state ϵ−machine, there are two routes from the non-troll state to the troll state in the average three-state ϵ−transducer; one is directly from the non-troll state to the troll state, and the other is through the transition state. Also, a user can switch from the transition state to both troll and non-troll states. In other words, when only relying on their own input, users enter the transition state before becoming trolls and will revert to non-trolling from there. However, when influenced by social input, users can switch from non-trolling to trolling directly (responding to a troll post, 1|1: 0.29) and get stuck going back and forth between trolling and the transition state.

Furthermore, comparing the average three-state ϵ−machine and ϵ−transducer, we noticed that the self-motived switch from the non-troll via transaction to troll state has a probability of 12.48% in ϵ−machines, and that in ϵ−transducer it is 10.5%. The Welch’s *t*-test showed no significant difference between the two (*t* = 1.88, *df* = 22, *p* = 0.074), indicating the information derived from one’s own past behavior is the same in self- and social-induced models. The probability of the socially motivated switch from non-troll state directly to troll state is 29% (responding to a troll post 1|1), which is almost three times bigger than the self-motivated switch in the social-induced model, indicating more information provided from social input.

We also conducted an additional investigation into average machines with more than three states, which revealed that states are indistinctive in those machines, unlike two- and three-state machines, whose states can be easily identified as the troll, non-troll, and transaction state. Some states in the multiple-state machine have only one option of switching in and out, leading to a specific flow among states. This once again illustrated that trolling behavior is a complex behavior with various shades of gray that are embedded in complicated internal situations and external environments.

## 5. Discussion

We derived the unique, minimally complex, maximally predictive model to decompose the dynamics of trolling behaviors. As one of the most concerning online deviant behaviors, trolling is known to irritate Internet users, disrupt online discussions, and harm online communities [3,61,72]. By approaching from a behavioral scientific perspective and dynamic system theory, we examined the hidden patterns of the behavioral models. The hidden pattern shows that trolling behavior is a complex process that can be self-driven and socially induced, indicating internal and external influences for people engaging in trolling behaviors. This interplay between individual tendencies and contextual triggers reflects the early roots of trolling research in personality psychology, which has highlighted how traits such as psychopathy and narcissism [27] as well as emotional reactivity and perceived social cues [28] contribute to trolling behavior. These findings reinforce the view that trolling is not solely a product of platform dynamics or message content but is shaped by the interplay between psychological dispositions and real-time social interactions.

One important finding of this study is that seeing trolling behaviors will encourage individuals to engage in trolling behaviors and reinforce the engagement. Consistent with previous research [26,55], the complex interpersonal dynamics of trolling behavior revealed that trolling is contingent on the ongoing and simultaneous flow of communicative interactions with others. Just like a good part of a traffic jam is produced by the reactions of drivers to others, here, a good part of additional trolling behavior is produced by users reacting to and interacting with trolls. One possible explanation is the online disinhibition effect with the anonymous environment of Reddit and negative emotional contagion. Unlike other popular social media platforms such as Facebook or Twitter, Reddit encourages individuals not to use their real identities [58]. Anonymity serves the purpose of physical, virtual, and emotional distancing between individuals, leading to toxic disinhibition [73]. The negative emotional contagion fuels the process of disinhibition. Individuals are likely to be influenced by the contextual environment and the one they interact with.

Our findings also reveal a notable speed in the spread of trolling behavior. Specifically, we observed that users can transition from a non-trolling or transit state into a trolling state after a single exposure to another user’s trolling behavior, and that the probability of such a transition is almost three times as high as the probability of entering a trolling state based on self-driven dynamics alone. This finding supports prior research suggesting that trolling, unlike cyberbullying, may require minimal exposure to escalate [24]. In addition, we observed patterns of behavioral variability in these state transitions, suggesting a degree of stochasticity in the emergence and resolution of trolling behavior. These findings highlight trolling as both a socially contagious and contextually sensitive behavior, driven by a combination of external inputs and dynamic internal responses.

Furthermore, we unexpectedly found that activeness was not related to trolling behaviors. In contrast to growing concerns on the normalization of online deviant behaviors, we did not find that active users who spent more time on the Internet were trolls more. Unlike the gaming environment [55], which requires an immediate response, online discussions may offer an opportunity for individuals to revisit their messages, deliberate on their thoughts, and process others’ messages. This thinking process can calm down negative emotions in the short term and can help develop critical thinking ability and accumulate knowledge in the long run. By actively engaging in online discussions, individuals are likely to become more informed decision-makers regarding trolling behaviors. This finding may offer hope as regards civil online deliberative discussion.

A practical implication that can be derived from the study is the importance of individuals. Preventing deviant behaviors from contaminating online communities is not solely the responsibility of platform designers, policymakers, or community moderators. While it is widely recognized that platform policies and community norms on the aggregate level can tame trolls, the study hints at the critical role of individuals, who can easily be influenced and carry out trolling behaviors. In addition to strategies on the aggregate level to promote non-trolling norms, individual-level strategies are also essential to intervene in developing trolling and other online deviant behaviors. Even though trolling behavior is a complex process and hard to predict, different reaction strategies can be taken to jump out of the “trolling loop”. For instance, individuals can implement the idea of “Don’t feed the troll”, as responding emotionally to trolling increases the likelihood of becoming a troll yourself. Instead of responding with emotional outbursts immediately, individuals can be guided to critically analyze and reflect on the messages to see if they are worth replying to. In addition to educating individuals, the platform can also develop strategies to quickly identify potential trolls and trolling messages to take action to break the “trolling loop”. Therefore, efforts aimed at individual-level intervention should also work towards combating online deviant behaviors.

### Limitations and Future Work

A few limitations should be kept in mind when we interpret the findings of the study. First, our findings may not be generalizable to other social media platforms, such as YouTube or Twitter, as Reddit has its own culture, and it especially takes a different approach to controversial messages and communities. Their anti-harassment policy relies heavily on sub-reddit moderators, leading to a different treatment of the same message in different subreddits. This is also related to our second limitation on the definition of trolling. Trolling, or, broadly, online deviant behaviors, is a complex concept that people may perceive differently contextually, leading to various reactions reflected in our dataset. Third, while the BERT-based classifier performed well, we did not apply interpretability techniques, limiting insight into which features the model relied on. Future work could incorporate such methods to better understand the cues associated with trolling. Fourth, one important assumption is that our time series is a discrete-time stochastic process, while the duration between two time points may vary. The variation may further complicate the mathematical structures of trolling behaviors. Fifth, our treatment of missing data may also lead to potential errors, such as some removed non-trolling messages and deleted trolling messages that are miscategorized. Further studies may also take seasonality into consideration. Lastly, our average machine approach did not further cluster machines by convergence structure or behavioral subtype. As a result, the averaged representations may smooth over meaningful distinctions between subgroups of users. Future work could apply clustering techniques prior to averaging to better account for such variation. In addition, further studies can be conducted to investigate the multiple-state machines to better understand the difference between hidden states and the users’ trolling behaviors. Further studies can also focus on investigating how other variables, such as different emotions displayed in the comments, affect trolling behavior and stochastic models of the behavior.

Despite the limitations, this study offers a novel behavioral scientific approach to analyzing communication as a dynamic system, using predictive modeling to trace how trolling behaviors emerge and unfold over time. Our findings provide formal evidence that trolling is not merely an individual act but is often socially induced and contagious, with exposure significantly increasing the likelihood of engagement. These findings move beyond descriptive accounts to demonstrate the underlying dynamics that sustain trolling in online environments. Crucially, they suggest that cultural interventions, such as shifting group norms, amplifying constructive behavior, and interrupting cycles of negativity, may be key to addressing pervasive online deviance in ways that go beyond punitive or legislative responses.

## Figures and Tables

**Figure 1 entropy-27-00542-f001:**
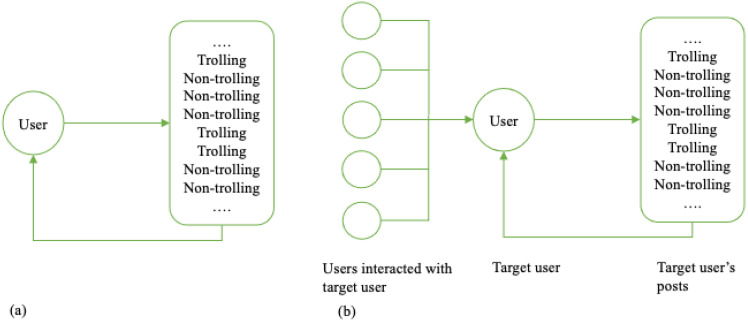
Self-driven and social-induced Models. *Note.* A schematic representation of the classes of models used. (**a**) The self-driven model, where the individual’s behavior only depends on their past behaviors. (**b**) The social-induced model, where the individual’s behavior is influenced by their social inputs and their own past behaviors.

**Figure 2 entropy-27-00542-f002:**
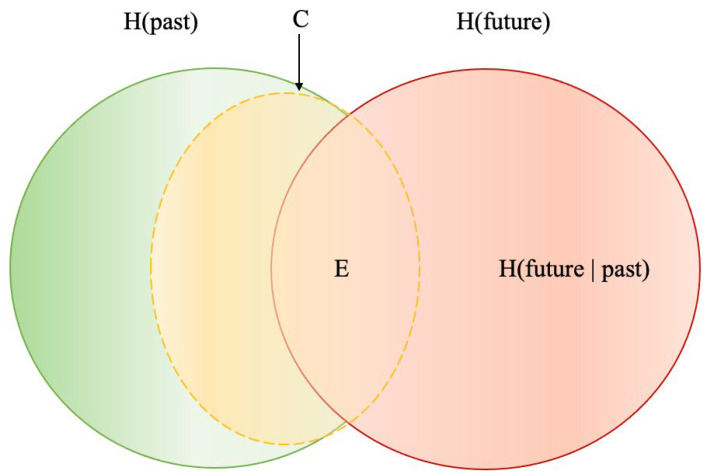
Venn diagram of stationary stochastic processes. *Note.* The green circle represents the information needed for past events, and the red circle represents the information needed for future events. As past events can inform what will happen in the future, the overlapped area of the circles represents the mutual information that the past transmits to the future, as predictable information E. The orange dashed ellipse represents the predictive complexity C. The area remaining in the red circle after subtracting the area of E is the conditional entropy $H\left(future\right|\;past)$, and the entropy rate h is the per-symbol rate of $H\left(future\right|\;past)$.

**Figure 3 entropy-27-00542-f003:**
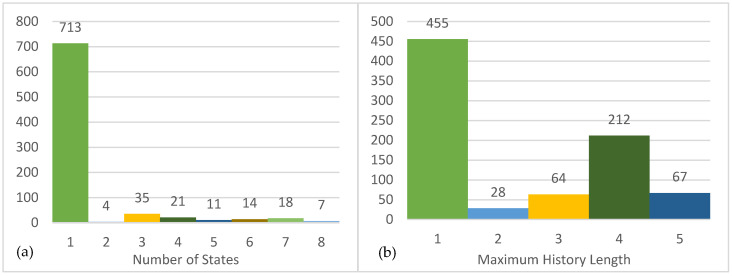
Distribution of states and maximum history length in the self-driven models. (**a**) The distribution of states of ϵ−machines. (**b**) The distribution of the maximum history length of ϵ−machines.

**Figure 4 entropy-27-00542-f004:**
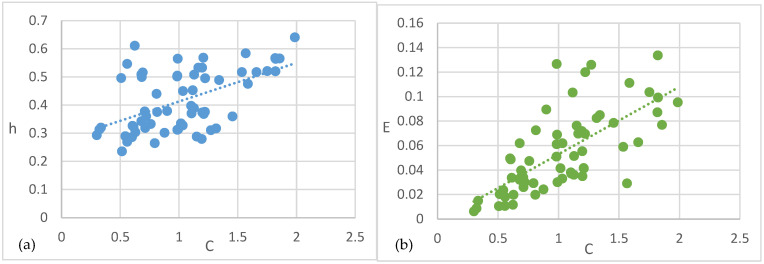
Predictive complexity vs. remaining uncertainty and predictable information with regression lines of self-driven models. (**a**) Predictive complexity vs. remaining uncertainty. (**b**) Predictive complexity vs. predictable information.

**Figure 5 entropy-27-00542-f005:**
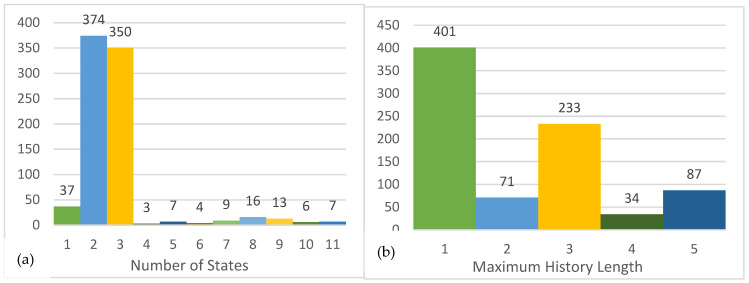
Distribution of states and maximum history length in the social-induced models. (**a**) The distribution of states of ϵ−machines. (**b**) The distribution of the maximum history length of ϵ−transducer.

**Figure 6 entropy-27-00542-f006:**
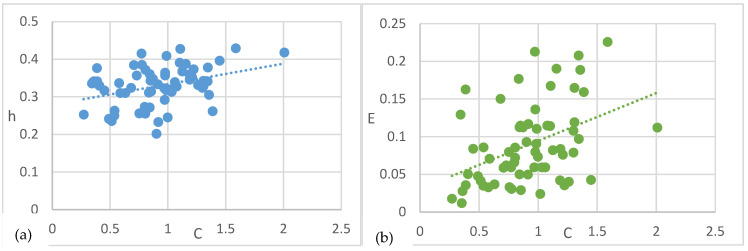
Predictive complexity vs. remaining uncertainty and predictable information with regression lines of social-induced models. (**a**) Predictive complexity vs. remaining uncertainty. (**b**) Predictive complexity vs. predictable information.

**Figure 7 entropy-27-00542-f007:**
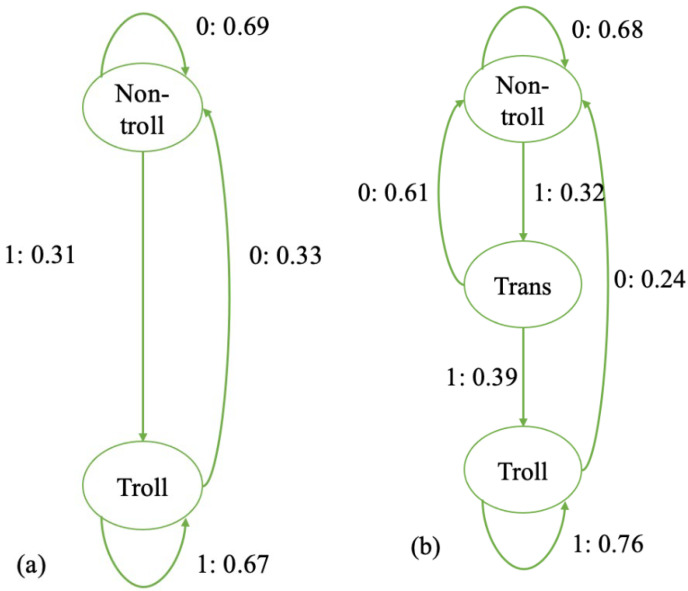
The average two- and three-state ϵ−machines for users’ trolling behavior. (**a**) The average two-state machine (N = 4); (**b**) the average three-state machine (N = 35). The number before the colon indicates whether a user trolls or not, and the number after the colon is the probability of moving from one state to another.

**Figure 8 entropy-27-00542-f008:**
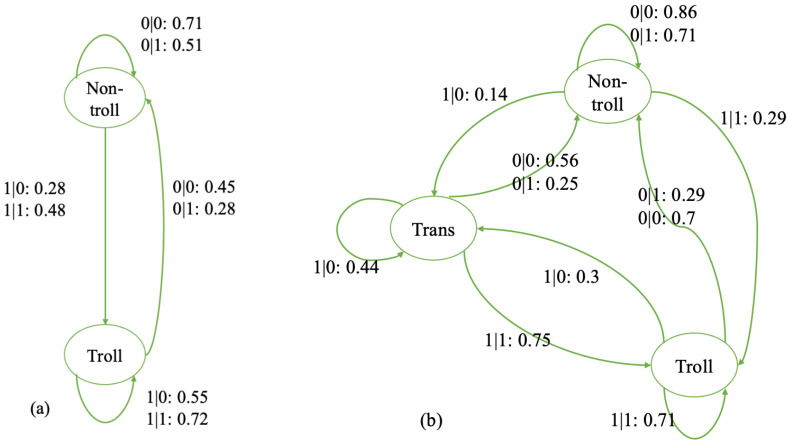
The average two- and three-state ϵ−transducers for the users’ trolling behavior. (**a**) The average two-state machine (N = 374); (**b**) the average three-state machine (N = 350). The number before the colon indicates whether the user’s behavior and the social input are trolling or not, and the number after the colon is the probability of moving from one state to another.

**Table 1 entropy-27-00542-t001:** ANOVA results for the predictable information, predictive complexity, and remaining uncertainty of self-driven models.

	Effect	F	
C	Number of States	17.973	***
	$L_{max}$	3.539	
E	Number of States	0.003	
	$L_{max}$	1.878	
h	Number of States	20.198	***
	$L_{max}$	8.405	**

*Note.* ‘***’ 0.001 ‘**’ 0.01 ‘*’ 0.05 ‘.’ 0.1 ‘ ’ 1.

**Table 2 entropy-27-00542-t002:** ANOVA results for the predictable information, predictive complexity, and remaining uncertainty of social-induced models.

	Effect	F	
C	Number of States	61.016	***
	$L_{max}$	19.442	***
E	Number of States	21.853	***
	$L_{max}$	5.265	*
h	Number of States	79.270	***
	$L_{max}$	28.973	***

*Note.* ‘***’ 0.001 ‘**’ 0.01 ‘*’ 0.05 ‘.’ 0.1 ‘ ’ 1.

## Data Availability

The data presented in this study are available on request from the corresponding author due to privacy and ethical restrictions.

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
