# Peer review of "Will You Become the Next Troll? A Computational Mechanics Approach to the Contagion of Trolling Behavior"

_entropy, 2025, doi:10.3390/e27050542_

Round 1
Reviewer 1 Report
Comments and Suggestions for Authors
For the authors’ convenience, I have (roughly) split up my review according to the headings used by the authors themselves in their paper. Please note that I am not a computer scientist, so please take my feedback with a grain of salt, as I am coming at this article from the angle of a communication specialist who researches trolling. I also review as I read, so if I mention something earlier that you mention later in the article, simply take it as a sign that it might be helpful to the reader to place it earlier in the piece.
Introduction: I think the authors have covered a lot of the important work here, and I applaud them for taking the time to engage with the trolling research community’s work. On the whole, I don’t disagree with what the authors are saying with the exception of page 1, lines 40-41: if it talks like a troll, it is a troll. Although the authors are correct in saying that this is the majority view, there is at least one article that goes against this viewpoint and argues that the metadata associated with trolls is more important than the linguistic content. It can be found at the following doi: https://doi.org/10.1093/jcmc/zmz014. I would also like to point out that trolling is, as noted by the researchers, incredibly broad. Although it is perhaps not typical to put an operationalization in the introduction, given the breadth of behaviours that the authors could be referring to with the word “trolling”, I would strongly recommend making a statement as early as possible in the article about what THIS article considers to be trolling and what THIS article will be tackling, as that will heavily impact the article’s validity and usefulness to whom.
Trolling as a Communication Dynamic Process: I thoroughly enjoyed reading this literature review! However, I do think it’s a bit misleading to say that trolling has been studied mostly from a linguistic perspective. Although there are many fantastic linguists who do indeed focus on trolling, Claire Hardaker being a cited favourite, the beginnings of empirical work on trolling specifically actually come from personality psychology, or the psychology of individual differences (see https://doi.org/10.1016/j.paid.2014.01.016). The work of Evita March, in particular, stands out as especially influential. Although the way the authors have chosen to study trolling spread is indeed important and novel, I would like to see some nod to the forebears in other disciplines, as communication and computational work are only two of the disciplines that study this phenomenon. In addition, there has been some empirical evidence within the world of trolling research that there is a trolling cycle and that trolling is, to some degree, contagious (see the earlier mentioned article about ‘talking like a troll’ for one example, although there are more out there). It might be worth mentioning some of these examples as further evidence beyond pure theory that trolling is, in all likelihood, contagious.
Predictive State Models: To claim I understood this section perfectly would be lying, but I do have one particular qualm after reading the full theoretical background. What are the exact differences between RQs 1, 2, and 3? From what I understood from this section, no algorithm is capable of “seeing” the hidden states that could be driving trolling (i.e., psychological states and traits), and so your model is based on previous interactions with what have been labeled “trolls” as a predictor. If so, then what is the difference between social influence (RQ1), contagion (RQ2), and hidden patterns (RQ3)? This could be me having a misunderstanding, but I just want to clarify these terms.
Methods: What is the “Google Big Query”? I’ve never heard this term before. Also, my genuine sympathy for those poor seven undergrads. I hope they were well-compensated in some way. Other than that, my only question pertains to data-cleaning. Machine-learning is notoriously bad at things like sarcasm and irony, which I can see that you have valiantly battled using human coding for the training system, but are there any data cleaning procedures that were undertaken before inputting the information? Perhaps this could be added in supplementary materials or an appendix if it is available. I am also assuming that this a ‘black box technique’ in the sense that we don’t actually know which features the system is using to determine what is trolling and non-trolling? Perhaps I will learn in the results section, but this could be a limitation worth mentioning later on if this is indeed the case.
Results: Reading this now, I think I understand the differences between the RQ’s better now, but I would encourage the authors to explain them more thoroughly earlier so that we non-computational readers don’t give up halfway through. The results are fascinating, and definitely go both with and against existing literature on the idea of contagion in trolling research. The idea of things happening very quickly is supported in a lot of the qualitative work around trolling (as opposed to cyberbullying, which usually requires multiple exposures to turn victim to perpetrator), but the randomness is an interesting factor I haven’t seen in existing work. The interaction between internal and external factors harks back to the personality psychology work I mentioned earlier, both its strengths and weaknesses, so I would recommend even more strongly to add in some of that work to better tie in these results to extant literature.
Discussion: This whole article was a great read, and I thoroughly agree with the conclusions drawn here. Engage a little bit more with a slightly wider variety of trolling work in this discussion, and I will be happy to endorse publication and ideally cite this myself in later works!
Author Response
|
Comments 1: For the authors’ convenience, I have (roughly) split up my review according to the headings used by the authors themselves in their paper. Please note that I am not a computer scientist, so please take my feedback with a grain of salt, as I am coming at this article from the angle of a communication specialist who researches trolling. I also review as I read, so if I mention something earlier that you mention later in the article, simply take it as a sign that it might be helpful to the reader to place it earlier in the piece. |
|
Response 1: We thank the reviewer for the thoughtful and detailed feedback provided throughout the review. We greatly appreciate the insights offered from a communication studies perspective, especially in relation to trolling research. We also note the reviewer’s point about the organization of the article and have taken this into account when revising the structure and flow to better guide readers across disciplines. |
|
Comments 2: Introduction: I think the authors have covered a lot of the important work here, and I applaud them for taking the time to engage with the trolling research community’s work. On the whole, I don’t disagree with what the authors are saying with the exception of page 1, lines 40-41: if it talks like a troll, it is a troll. Although the authors are correct in saying that this is the majority view, there is at least one article that goes against this viewpoint and argues that the metadata associated with trolls is more important than the linguistic content. It can be found at the following doi: https://doi.org/10.1093/jcmc/zmz014. I would also like to point out that trolling is, as noted by the researchers, incredibly broad. Although it is perhaps not typical to put an operationalization in the introduction, given the breadth of behaviours that the authors could be referring to with the word “trolling”, I would strongly recommend making a statement as early as possible in the article about what THIS article considers to be trolling and what THIS article will be tackling, as that will heavily impact the article’s validity and usefulness to whom. |
|
Response 2: We thank the reviewer for their thoughtful comments and generous engagement with our introduction. In response to the reviewer’s helpful suggestion, we revised lines 41-46 (which are highlighted) to explicitly acknowledge additional scholarship that goes beyond the linguistic features of trolling. We have also revised and rearranged the subsequent paragraph to clearly define what we mean by “trolling” in this study (lines 47-50, highlighted). We agree that an explicit definition early in the article is crucial given the broad and contested nature of the term, and we hope this revision improves the clarity and usefulness of our framing. |
|
Comments 3: Trolling as a Communication Dynamic Process: I thoroughly enjoyed reading this literature review! However, I do think it’s a bit misleading to say that trolling has been studied mostly from a linguistic perspective. Although there are many fantastic linguists who do indeed focus on trolling, Claire Hardaker being a cited favourite, the beginnings of empirical work on trolling specifically actually come from personality psychology, or the psychology of individual differences (see https://doi.org/10.1016/j.paid.2014.01.016). The work of Evita March, in particular, stands out as especially influential. Although the way the authors have chosen to study trolling spread is indeed important and novel, I would like to see some nod to the forebears in other disciplines, as communication and computational work are only two of the disciplines that study this phenomenon. In addition, there has been some empirical evidence within the world of trolling research that there is a trolling cycle and that trolling is, to some degree, contagious (see the earlier mentioned article about ‘talking like a troll’ for one example, although there are more out there). It might be worth mentioning some of these examples as further evidence beyond pure theory that trolling is, in all likelihood, contagious. |
|
Response 3: We sincerely thank the reviewer for this generous and insightful comment, and we appreciate the opportunity to clarify and strengthen our literature review. To avoid the misleading impression that trolling has been studied predominantly from a linguistic perspective, we revised the phrasing in our discussion of definitional approaches to trolling (line 98-101, highlighted). We now explicitly acknowledge the foundational role of personality psychology in early empirical work on trolling (line 101-104, highlighted). In addition, we have expanded our discussion to incorporate empirical evidence of trolling contagion (line 174-179, highlighted). We believe these additions more accurately reflect the interdisciplinary foundations of trolling research and support the framing of trolling as a dynamic and socially embedded phenomenon. |
|
Comments 4: Predictive State Models: To claim I understood this section perfectly would be lying, but I do have one particular qualm after reading the full theoretical background. What are the exact differences between RQs 1, 2, and 3? From what I understood from this section, no algorithm is capable of “seeing” the hidden states that could be driving trolling (i.e., psychological states and traits), and so your model is based on previous interactions with what have been labeled “trolls” as a predictor. If so, then what is the difference between social influence (RQ1), contagion (RQ2), and hidden patterns (RQ3)? This could be me having a misunderstanding, but I just want to clarify these terms. |
|
Response 4: We thank the reviewer for raising this important point. We recognize that the distinctions between RQ1, RQ2, and RQ3 were not sufficiently clear in the earlier version of the manuscript, and we have revised the literature review and theoretical framing sections to clarify these differences. Specifically, RQ1 asks whether an individual’s trolling behavior is better predicted by their own past behavior or by social inputs from others. This question is meant to disentangle self-history dependence from social influence assessing whether trolling is more self-driven or socially influenced (line 139-141, highlighted). In contrast, RQ2 focuses more narrowly on prior trolling by others. We tested whether exposure to trolling behavior specifically increases the probability that a user will engage in trolling, pointing to possible behavioral contagion mechanisms (line 182-184, highlighted). To address the reviewer’s helpful suggestion, we have also revised RQ3 to more explicitly reflect what we are investigating (line 229-230, highlighted). Originally framed as a search for “hidden patterns,” this question aims to explore whether user behavior exhibits structured transitions, such as remaining in a trolling state after initial engagement, or showing signs of return to non-trolling participation, based on reconstructed sequences. We are not claiming access to latent psychological states but rather asking whether patterns in observable behavior (e.g., sustained trolling once it begins or transitional states) suggest underlying dynamics that unfold over time (line 218-224, highlighted). We hope these revisions clarify the conceptual distinctions among the three research questions and how each contributes to understanding trolling as a dynamic and socially embedded behavior. |
|
Comments 5: Methods: What is the “Google Big Query”? I’ve never heard this term before. Also, my genuine sympathy for those poor seven undergrads. I hope they were well-compensated in some way. Other than that, my only question pertains to data-cleaning. Machine-learning is notoriously bad at things like sarcasm and irony, which I can see that you have valiantly battled using human coding for the training system, but are there any data cleaning procedures that were undertaken before inputting the information? Perhaps this could be added in supplementary materials or an appendix if it is available. I am also assuming that this a ‘black box technique’ in the sense that we don’t actually know which features the system is using to determine what is trolling and non-trolling? Perhaps I will learn in the results section, but this could be a limitation worth mentioning later on if this is indeed the case. |
|
Response 5: We thank the reviewer for this thoughtful and detailed comment. We have clarified in the revised manuscript that Google BigQuery is a large-scale cloud-based data warehouse that enables querying massive datasets using SQL-like syntax. This clarification has been added to the methods section for readers unfamiliar with the platform (line 272-273, highlighted). Regarding the data preprocessing, in the revised methods section, we now specify that we conducted minimal text preprocessing, including whitespace normalization and removal of special characters, emojis, and URLs (line 326-328, highlighted). This decision was based on the fact that BERT-based models are pre-trained on unprocessed web text and can handle informal and noisy data reasonably well. Finally, we have added a note to the limitations section acknowledging the model’s opacity (line 716-718, highlighted). While the BERT-based classifier achieved good performance, we did not employ interpretability techniques in this study, and we now highlight this as a potential limitation that future work could address. |
|
Comments 6: Results: Reading this now, I think I understand the differences between the RQ’s better now, but I would encourage the authors to explain them more thoroughly earlier so that we non-computational readers don’t give up halfway through. The results are fascinating, and definitely go both with and against existing literature on the idea of contagion in trolling research. The idea of things happening very quickly is supported in a lot of the qualitative work around trolling (as opposed to cyberbullying, which usually requires multiple exposures to turn victim to perpetrator), but the randomness is an interesting factor I haven’t seen in existing work. The interaction between internal and external factors harks back to the personality psychology work I mentioned earlier, both its strengths and weaknesses, so I would recommend even more strongly to add in some of that work to better tie in these results to extant literature. |
|
Response 6: We thank the reviewer for this generous and thoughtful feedback. In response, we have revised the earlier sections to more clearly articulate the distinctions between RQ1, RQ2, and RQ3, which have been detailed in Response 4. We hope this clearer formulation helps convey the distinct analytical contribution of this research question. We also appreciate the reviewer’s insightful observation about the rapid onset and apparent randomness of trolling behaviors, and we have incorporated language in the result discussion to emphasize how our findings both echo and extend existing qualitative research on trolling dynamics (line 669-679, highlighted). Lastly, we elaborated on the connection between our findings and earlier personality psychology research, highlighting how the observed interaction between internal and external influences reflects foundational work on traits and situational factors in trolling (line 648-654, highlighted). We hope these additions improve the interpretive depth of the discussion and further demonstrate how our findings contribute to and expand upon existing literature. |
|
Comments 7: Discussion: This whole article was a great read, and I thoroughly agree with the conclusions drawn here. Engage a little bit more with a slightly wider variety of trolling work in this discussion, and I will be happy to endorse publication and ideally cite this myself in later works! |
|
Response 7: We are truly grateful for the reviewer’s generous comments and encouragement. In response to the suggestion, we have expanded the discussion to incorporate a broader range of literature on trolling by including work from both qualitative and psychological traditions, to better contextualize our findings and highlight their relevance across disciplinary perspectives. We hope these additions strengthen the manuscript’s contribution and demonstrate its potential to engage with and complement existing work in the field. |
Reviewer 2 Report
Comments and Suggestions for Authors
Review of “Will You Become the Next Troll? A Computational Mechanics
Approach to the Contagion of Trolling Behavior”, Entropy manuscript 3551885.
Summary: This paper studies trolling on Reddit using the computational mechanics framework developed by Crutchfield and colleagues. The data studied were approximately 13 million Reddit posts from a total of 16,000 users during 2016 and 2017. The authors had 7 undergraduates evaluate a small subset (0.12%) of the posts as trolling or non-trolling based on a protocol, and then they used these human generated annotations to train and a test a BURT model, finally using the BURT model to classify the rest of the 13 million. The data were then turned into a directed graph indicating the user interactions across time with edges of type self -> self and other -> self, each node being labeled with its training status. Then the researchers used Darmon (2015)’s Causal State Splitting Reconstruction (CSSR) model, based on Shalizi & Crutchfield (2001)’s framework, to construct Markov models of individual’s causal states and state transition probabilities, considering two possible model classes: (i) an individual’s future behavior depends only on their past behavior (“self-driven model”); (ii) an individual’s future behavior depends on both their past behavior and the behaviors of people whose Reddit posts they were exposed to (“social-induced model”) They studied the resulting models to answer three research questions:
RQ1: Is trolling socially influenced?
RQ2: Is trolling behavior contagious?
RQ3: Is there any hidden pattern of trolling behavior?
The authors reported evidence for a positive answer to all three questions. One overall conclusion is “trolling behavior is a complex behavior with various shades of grey that are embedded in complicated internal situations and external environments.” (ll. 620-621).
Critique: I find the overall undertaking of this paper interesting and inspiring. I want to support the authors in their creative and constructive use of an approach to modeling which I feel has not been explored nearly as much its potential warrants. With that positive note in mind, I raise some concerns about the significance of the results as reported. If these issues can be addressed, then I think the work will be worth publishing.
My take is:
- (1) The main results reported in Section 3.3 on RQ1 and RQ2 are reasonably compelling (Welch’s tests for significance of social influence and contagious spread of trolling). These two hypotheses are rather generally believed, so it’s not a surprise to confirm them, but it is valuable to have a rigorous formal method of doing so.
- (2) I find RQ3 (“hidden pattern”) more problematic---it’s not a very specific claim so it’s hard to say what counts as strong evidence in its favor. The argument in support of it seems to be based partly on average state-machines generated for 2-state and 3-state reconstructions. The average state-machines concern me as I explain in more detail below.
- (3) The overall conclusion mentioned above is pretty tepid---it doesn’t say much beyond what an observant person might reasonablu surmise from a casual look at the scene.
- (4) The demonstrations of relations between information-theoretic quantities are not fully compelling.
- (5) The paper is verbose---the writing would best be made more succinct.
Regarding (2) [on RQ3---existence of “a hidden pattern in trolling behavior”], I recommend that the authors either drop the question or refine it. The intuitive question seems too open-ended. If a pattern can be any structure at all, then it’s not very informative to get a yes answer. I think the authors might mean something like, “Does Causal State Machine reconstruction yield machines with more than one state?” If this is the question, I recommend stating it explicitly and also saying why this is an important question---how would a positive answer give us insight beyond the RQ1/RQ2 results?
Secondly, also regarding (2), the argument for hidden patterns seems partly based on Section 3.4 “Average Machines”. The authors report “simple averages” of machines that are in the same model class (self-driven vs. social-induced), have more than 1 states, and have the same number of states. The authors should describe formally exactly how they did the averaging. My best guess is: they selected a 1-1 correspondence between the states of the two machines; they treated missing transitions as transitions with 0 probability; they averaged the probabilities across the aligned machines. This procedure is of questionable usefulness for several reasons: (i) it might be that the machines fall into highly distinct and uniform subtypes; if so, the average machine may not resemble the individual machines closely at all so, for example, the putative “transition state” could be instead a trollhood-forswearing state for some users. (ii) it’s not clear how the states are aligned. Maybe Darmon (2015) explains this, but I recommend that the authors summarize the principle and given a citation if so.
Regarding (3), the bland overall conclusion, I recommend dropping this conclusion and focusing on the value of demonstrating positive answers to RQ1 and RQ2, possibly brainstorming how these might point to ways that cultural (as opposed to legislative) transcendence of pervasive trolling might come about.
Regarding (4), relations between information-theoretic quantities, there are some concerning properties of Figures 4a and 6a. First, the rightmost data point of Figure 4a seems to have a C value around 2.4. If I understood correctly, C is the average entropy of the reconstructed machine’s states. The authors state previously that the maximal number of reconstructed machine states for the self-driven model was 8. An 8-state Markov process can have a maximum individual state entropy of 2.08 (entropy of the uniform distribution on 8 states). Thus, it should not be possible for the average entropy over a large sample of successive states to be greater than 2.08.
Second, regarding Figures 4a and 6a, the results seem likely to be strongly driven by the left most data point: 0 complexity with 0 remaining uncertainty. If this point were taken away, it looks like the correlation among the data points would be considerably weakened. Presumably, this left-most state corresponds to the single-state model reconstructions. Given the definition of causal states, its impossible for a model with a higher C value to have 0 average remaining entropy---since entropy is nonnegative, the only way for the average entropy to be 0 is for all the states to have 0 entropy, in which case C is not greater than 0, contradicting the assumption. Therefore, the positive correlation in Figure 4a is, in a sense inevitable, under the very weak assumption that there is some uncertainty in some model. Thus, the comment in the narrative about a general positive relationship between complexity C and remaining uncertainty (ll 448-450) does not seem strongly supported by the data. If the leftmost point were removed and the correlation still came out significant, I would be more convinced. Similar considerations apply to Figure 6a.
Regarding (5), verbosity, an example is: “The number of states indicates the size of the machine. The more states there are, the larger the machine is.” (One of these sentences would do). I recommend striving to shorten the narrative by at least 2 pages.
Author Response
|
Comments 1: Critique: I find the overall undertaking of this paper interesting and inspiring. I want to support the authors in their creative and constructive use of an approach to modeling which I feel has not been explored nearly as much its potential warrants. With that positive note in mind, I raise some concerns about the significance of the results as reported. If these issues can be addressed, then I think the work will be worth publishing.
|
|
Response 1: We are deeply grateful to the reviewer for their generous and encouraging feedback on our study. We particularly appreciate the recognition of our modeling approach and the potential it holds for studying online behavioral dynamics, which aligns closely with our motivation for pursuing this work. In response to the reviewer’s concern about the significance of the results, we have made several important revisions to improve the clarity and impact of our findings. Specifically, we (1) provided more detailed interpretations of key results, (2) added contextual framing to highlight their broader implications, and (3) clarified how our modeling outputs offer new insights beyond prior work on trolling behavior. We hope these revisions strengthen the manuscript and help demonstrate the contribution of this work to both theoretical understanding and methodological innovation in the study of online deviance. |
|
Comments 2: (1) The main results reported in Section 3.3 on RQ1 and RQ2 are reasonably compelling (Welch’s tests for significance of social influence and contagious spread of trolling). These two hypotheses are rather generally believed, so it’s not a surprise to confirm them, but it is valuable to have a rigorous formal method of doing so. |
|
Response 2: We sincerely thank the reviewer for this generous and thoughtful comment. We are glad the reviewer found the results compelling and appreciate the recognition of our effort to apply a formal, rigorous method to empirically validate widely held assumptions about social influence and contagion in trolling behavior. |
|
Comments 3: (2) I find RQ3 (“hidden pattern”) more problematic---it’s not a very specific claim so it’s hard to say what counts as strong evidence in its favor. The argument in support of it seems to be based partly on average state-machines generated for 2-state and 3-state reconstructions. The average state-machines concern me as I explain in more detail below. Regarding (2) [on RQ3---existence of “a hidden pattern in trolling behavior”], I recommend that the authors either drop the question or refine it. The intuitive question seems too open-ended. If a pattern can be any structure at all, then it’s not very informative to get a yes answer. I think the authors might mean something like, “Does Causal State Machine reconstruction yield machines with more than one state?” If this is the question, I recommend stating it explicitly and also saying why this is an important question---how would a positive answer give us insight beyond the RQ1/RQ2 results? Secondly, also regarding (2), the argument for hidden patterns seems partly based on Section 3.4 “Average Machines”. The authors report “simple averages” of machines that are in the same model class (self-driven vs. social-induced), have more than 1 states, and have the same number of states. The authors should describe formally exactly how they did the averaging. My best guess is: they selected a 1-1 correspondence between the states of the two machines; they treated missing transitions as transitions with 0 probability; they averaged the probabilities across the aligned machines. This procedure is of questionable usefulness for several reasons: (i) it might be that the machines fall into highly distinct and uniform subtypes; if so, the average machine may not resemble the individual machines closely at all so, for example, the putative “transition state” could be instead a trollhood-forswearing state for some users. (ii) it’s not clear how the states are aligned. Maybe Darmon (2015) explains this, but I recommend that the authors summarize the principle and given a citation if so.
|
|
Response 3: We thank the reviewer for this thoughtful and important suggestion. (1) In response, we have substantially revised the framing of RQ3 to make both its meaning and its significance more explicit. We acknowledge that the original phrasing was too vague and may have seemed open-ended or underspecified. In the revised version, RQ3 is now framed as follows: We have also revised the manuscript text (line 218-224, highlighted) to emphasize why this question is important. Specifically, it allows us to move beyond the identification of external influences and toward an understanding of trolling as a potentially self-sustaining or temporally structured behavior. This perspective is important for understanding the stickiness of trolling and how trolling may evolve even in the absence of immediate social cues. (2) We have also added a paragraph to the revised manuscript (line 566-577) that formally explains our procedure for constructing average machines. This includes a detailed description of how we aligned states across machines with the same number of states, following the state-ordering method. We also appreciated the reviewer’s concern regarding the potential heterogeneity of machines. While we grouped machines by model class and state count to ensure comparability, we did not further cluster them by convergence structure or behavioral subtype. We now acknowledge this in the manuscript as a limitation, as averaging across structurally diverse machines may obscure meaningful behavioral distinctions. We suggest that future work could explore clustering approaches prior to averaging to better account for such variation (line 724-728). |
|
Comments 4: (3) The overall conclusion mentioned above is pretty tepid---it doesn’t say much beyond what an observant person might reasonablu surmise from a casual look at the scene. Regarding (3), the bland overall conclusion, I recommend dropping this conclusion and focusing on the value of demonstrating positive answers to RQ1 and RQ2, possibly brainstorming how these might point to ways that cultural (as opposed to legislative) transcendence of pervasive trolling might come about. |
|
Response 4: We thank the reviewer for this helpful and constructive suggestion. In response, we have substantially revised the conclusion to better reflect the significance of our findings (line 733-742). Rather than ending with a broad generalization, the revised paragraph now emphasizes our key contributions and their implications for understanding trolling as a dynamic and socially driven process. We also incorporate the reviewer’s valuable recommendations to frame the impact of these findings in terms of cultural responses, such as norm-shaping and constructive social interventions, rather than legislative or punitive solutions. We believe this revised conclusion offers a stronger and more meaningful close to the paper. |
|
Comments 5: (4) The demonstrations of relations between information-theoretic quantities are not fully compelling. Regarding (4), relations between information-theoretic quantities, there are some concerning properties of Figures 4a and 6a. First, the rightmost data point of Figure 4a seems to have a C value around 2.4. If I understood correctly, C is the average entropy of the reconstructed machine’s states. The authors state previously that the maximal number of reconstructed machine states for the self-driven model was 8. An 8-state Markov process can have a maximum individual state entropy of 2.08 (entropy of the uniform distribution on 8 states). Thus, it should not be possible for the average entropy over a large sample of successive states to be greater than 2.08. Second, regarding Figures 4a and 6a, the results seem likely to be strongly driven by the left most data point: 0 complexity with 0 remaining uncertainty. If this point were taken away, it looks like the correlation among the data points would be considerably weakened. Presumably, this left-most state corresponds to the single-state model reconstructions. Given the definition of causal states, its impossible for a model with a higher C value to have 0 average remaining entropy---since entropy is nonnegative, the only way for the average entropy to be 0 is for all the states to have 0 entropy, in which case C is not greater than 0, contradicting the assumption. Therefore, the positive correlation in Figure 4a is, in a sense inevitable, under the very weak assumption that there is some uncertainty in some model. Thus, the comment in the narrative about a general positive relationship between complexity C and remaining uncertainty (ll 448-450) does not seem strongly supported by the data. If the leftmost point were removed and the correlation still came out significant, I would be more convinced. Similar considerations apply to Figure 6a. |
|
Response 5: We thank the reviewer for this careful and constructive observation. Upon review, we identified that the reported value for C in Figure 4(a) was the result of an error in the dataset, for some very specific values, rather than in the theoretical explanation. We have corrected this issue in the underlying data and updated the corresponding figures and results accordingly in the revised manuscript. The maximum value for C now falls within the expected range given the number of reconstructed machine states. Given that the mistake was only linked to some specific values, our overall finding does not change. This was an embarrassing data processing mistake, and we are very grateful the alert reviewer pointed this out! We now double-checked the data and are confident that the error in the data presentation was removed. In response to the reviewer’s second point, we re-ran the correlation analyses for Figure 4(a), 4(b), 6(a), and 6(b), this time excluding both the leftmost boundary point (C=0, h=0, or E=0) and statistical outliers. In all four cases, the positive correlations remained statistically significant, supporting the robustness of our findings. We now report these results in the updated results section. We appreciate the reviewer’s close attention, which helped us improve both the accuracy and interpretability of this part of the paper. |
|
Response to Comments on the Quality of English Language |
|
Point 1: (5) The paper is verbose---the writing would best be made more succinct. Regarding (5), verbosity, an example is: “The number of states indicates the size of the machine. The more states there are, the larger the machine is.” (One of these sentences would do). I recommend striving to shorten the narrative by at least 2 pages. |
|
Response 1: We thank the reviewer for this valuable suggestion. In the revised manuscript, we carefully reviewed the text for redundancy and have removed or condensed several passages to improve conciseness. At the same time, we have intentionally retained some explanatory language, particularly in the methods and theoretical framing sections to ensure clarity for a broader interdisciplinary readership, including those less familiar with computational modeling. Our goal was to strike a balance between accessibility and brevity, and we appreciate the reviewer’s guidance in helping us move toward a more streamlined narrative. |
Round 2
Reviewer 1 Report
Comments and Suggestions for Authors
The authors have fully satsified all of my requests. I am happy to endorse publication at this stage, and look forward to seeing more interdisciplinary papers like this in the future!